# Adv-BNN: Improved Adversarial Defense through Robust Bayesian Neural Network

**Xuanqing Liu[1], Yao Li[2,*], Chongruo Wu[3,*] & Cho-Jui Hsieh[1]**

[1]: Department of Computer Science, UCLA
Los Angeles, CA 90095, UCLA
{xqliu,choheish}@cs.ucla.edu
[2]: Department of Statistics, UC Davis
[3]: Department of Computer Science, UC Davis
Davis, CA 95616, USA
{crwu,yaoli}@ucdavis.edu

## Abstract

We present a new algorithm to train a robust neural network against adversarial attacks. Our algorithm is motivated by the following two ideas. First, although recent work has demonstrated that fusing randomness can improve the robustness of neural networks (Liu et al., 2017), we noticed that adding noise blindly to all the layers is not the optimal way to incorporate randomness. Instead, we model randomness under the framework of Bayesian Neural Network (BNN) to formally learn the posterior distribution of models in a scalable way. Second, we formulate the mini-max problem in BNN to learn the best model distribution under adversarial attacks, leading to an adversarial-trained Bayesian neural network. Experiment results demonstrate that the proposed algorithm achieves state-of-the-art performance under strong attacks. On CIFAR-10 with VGG network, our model leads to 14% accuracy improvement compared with adversarial training (Madry et al., 2017) and random self-ensemble (Liu et al., 2017) under PGD attack with 0.035 distortion, and the gap becomes even larger on a subset of ImageNet[1].

## 1 Introduction

Deep neural networks have demonstrated state-of-the-art performances on many difficult machine learning tasks. Despite the fundamental breakthroughs in various tasks, deep neural networks have been shown to be utterly vulnerable to adversarial attacks (Szegedy et al., 2013; Goodfellow et al., 2015). Carefully crafted perturbations can be added to the inputs of the targeted model to drive the performances of deep neural networks to chance-level. In the context of image classification, these perturbations are imperceptible to human eyes but can change the prediction of the classification model to the wrong class. Algorithms seek to find such perturbations are denoted as adversarial attacks (Chen et al., 2018; Carlini & Wagner, 2017b; Papernot et al., 2017), and some attacks are still effective in the physical world (Kurakin et al., 2017; Evtimov et al., 2017). The inherent weakness of lacking robustness to adversarial examples for deep neural networks brings out security concerns, especially for security-sensitive applications which require strong reliability.

To defend from adversarial examples and improve the robustness of neural networks, many algorithms have been recently proposed (Papernot et al., 2016; Zantedeschi et al., 2017; Kurakin et al., 2017; Huang et al., 2015; Xu et al., 2015). Among them, there are two lines of work showing effective results on medium-sized data (e.g., CIFAR-10). The first line of work uses adversarial training to improve robustness, and the recent algorithm proposed in Madry et al. (2017) has been recognized as one of the most successful defenses, as shown in Athalye et al. (2018). The second line of work adds stochastic components in the neural network to hide gradient information from attackers. In the black-box setting, stochastic outputs can significantly increase query counts for attacks using

---

*Indicates equal contribution.

[1]Code for reproduction has been made available online at https://github.com/xuanqing94/BayesianDefense

finite-difference techniques (Chen et al., 2018; Ilyas et al., 2018), and even in the white-box setting the recent Random Self-Ensemble (RSE) approach proposed by Liu et al. (2017) achieves similar performance to Madry's adversarial training algorithm.

In this paper, we propose a new defense algorithm called Adv-BNN. The idea is to combine adversarial training and Bayesian network, although trying BNNs in adversarial attacks is not new (e.g. (Li & Gal, 2017; Feinman et al., 2017; Smith & Gal, 2018)), and very recently Ye & Zhu (2018) also tried to combine Bayesian learning with adversarial training, this is the first time we scale the problem to complex data and our approach achieves better robustness than previous defense methods. The contributions of this paper can be summarized below:

- Instead of adding randomness to the input of each layer (as what has been done in RSE), we directly assume all the weights in the network are stochastic and conduct training with techniques commonly used in Bayesian Neural Network (BNN).
- We propose a new mini-max formulation to combine adversarial training with BNN, and show the problem can be solved by alternating between projected gradient descent and SGD.
- We test the proposed Adv-BNN approach on CIFAR10, STL10 and ImageNet143 datasets, and show significant improvement over previous approaches including RSE and adversarial training.

**Notations** A neural network parameterized by weights $\boldsymbol{w} \in \mathbb{R}^d$ is denoted by $f(\boldsymbol{x}; \boldsymbol{w})$, where $\boldsymbol{x} \in \mathbb{R}^p$ is an input example and $y$ is the corresponding label, the training/testing dataset is $\mathcal{D}_{\mathrm{tr/te}}$ with size $N_{\mathrm{tr/te}}$ respectively. When necessary, we abuse $\mathcal{D}_{\mathrm{tr/te}}$ to define the empirical distributions, i.e. $\mathcal{D}_{\mathrm{tr/te}} = \frac{1}{N_{\mathrm{tr/te}}} \sum_{i=1}^{N_{\mathrm{tr/te}}} \delta(x_i) \delta(y_i)$, where $\delta(\cdot)$ is the Dirac delta function. $\boldsymbol{x}_o$ represents the original input and $\boldsymbol{x}^{\mathrm{adv}}$ denotes the adversarial example. The loss function is represented as $\ell\big(f(\boldsymbol{x}_i; \boldsymbol{w}), y_i\big)$, where $i$ is the index of the data point. Our approach works for any loss but we consider the cross-entropy loss in all the experiments. The adversarial perturbation is denoted as $\boldsymbol{\xi} \in \mathbb{R}^p$, and adversarial example is generated by $\boldsymbol{x}^{\mathrm{adv}} = \boldsymbol{x}_o + \boldsymbol{\xi}$. In this paper, we focus on the attack under norm constraint Madry et al. (2017), so that $\|\boldsymbol{\xi}\| \leq \gamma$. In order to align with the previous works, in the experiments we set the norm to $\|\cdot\|_\infty$. The Hadamard product is denoted as $\odot$.

## 2 BACKGROUNDS

### 2.1 ADVERSARIAL ATTACK AND DEFENSE

In this section, we summarize related works on adversarial attack and defense.

**Attack:** Most algorithms generate adversarial examples based on the gradient of loss function with respect to the inputs. For example, FGSM (Goodfellow et al., 2015) perturbs an example by the sign of gradient, and use a step size to control the $\ell_\infty$ norm of perturbation. Kurakin et al. (2017) proposes to run multiple iterations of FGSM. More recently, C&W attack Carlini & Wagner (2017a) formally poses attack as an optimization problem, and applies a gradient-based iterative solver to get an adversarial example. Both C&W attack and PGD attack (Madry et al., 2017) have been frequently used to benchmark the defense algorithms due to their effectiveness (Athalye et al., 2018). Throughout, we take the PGD attack as an example, largely following Madry et al. (2017).

The goal of PGD attack is to find adversarial examples in a $\gamma$-ball, which can be naturally formulated as the following objective function:

$$\max_{\|\boldsymbol{\xi}\|_\infty \leq \gamma} \ell(f(\boldsymbol{x}_o + \boldsymbol{\xi}; \boldsymbol{w}), y_o). \tag{1}$$

Starting from $\boldsymbol{x}^0 = \boldsymbol{x}_o$, PGD attack conducts projected gradient descent iteratively to update the adversarial example:

$$\boldsymbol{x}^{t+1} = \Pi_\gamma \left\{ \boldsymbol{x}^t + \alpha \cdot \mathrm{sign}\Big(\nabla_{\boldsymbol{x}} \ell\big(f(\boldsymbol{x}^t; \boldsymbol{w}), y_o\big)\Big) \right\}, \tag{2}$$

where $\Pi_\gamma$ is the projection to the set $\{\boldsymbol{x}| \ \|\boldsymbol{x} - \boldsymbol{x}_o\|_\infty \leq \gamma\}$. Although multi-step PGD iterations may not necessarily return the optimal adversarial examples, we decided to apply it in our experiments, following the previous work of (Madry et al., 2017). An advantage of PGD attack over C&W attack

is that it gives us a direct control of distortion by changing $\gamma$, while in C&W attack we can only do this indirectly via tuning the regularizer.

Since we are dealing with networks with random weights, we elaborate more on which strategy should attackers take to increase their success rate, and the details can be found in Athalye et al. (2018). In random neural networks, an attacker seeks a universal distortion $\boldsymbol{\xi}$ that cheats a majority of realizations of the random weights. This can be achieved by maximizing the loss expectation

$$\boldsymbol{\xi} \triangleq \underset{\|\boldsymbol{\xi}\|_\infty \leq \gamma}{\arg\max} \, \underset{\boldsymbol{w}}{\mathbb{E}}[\ell(f(\boldsymbol{x}_o + \boldsymbol{\xi}; \boldsymbol{w}), y_o)]. \tag{3}$$

Here the model weights $\boldsymbol{w}$ are considered as random vector following certain distributions. In fact, solving (3) to a saddle point can be done easily by performing multi-step (projected) SGD updates. This is done inherently in some iterative attacks such as C&W or PGD discussed above, where the only difference is that we sample new weights $\boldsymbol{w}$ at each iteration.

**Defense:** There are a large variety of defense methods proposed in recent years, e.g. denoiser based HGD (Liao et al., 2017) and randomized image preprocessing (Xie et al., 2017). Readers can find more from Kurakin et al. (2018). Below we select two representative ones that turn out to be effective to white box attacks. They are the major baselines in our experiments.

The first example is the adversarial training (Szegedy et al., 2013; Goodfellow et al., 2015). It is essentially a data augmentation method, which trains the deep neural networks on adversarial examples until the loss converges. Instead of searching for adversarial examples and adding them into the training data, Madry et al. (2017) proposed to incorporate the adversarial search inside the training process, by solving the following robust optimization problem:

$$\boldsymbol{w}^* = \underset{\boldsymbol{w}}{\arg\min} \, \underset{(\boldsymbol{x},y)\sim\mathcal{D}_{\mathrm{tr}}}{\mathbb{E}} \left\{ \max_{\|\boldsymbol{\xi}\|_\infty \leq \gamma} \ell\big(f(\boldsymbol{x} + \boldsymbol{\xi}; \boldsymbol{w}), y\big) \right\}, \tag{4}$$

where $\mathcal{D}_{\mathrm{tr}}$ is the training data distribution. The above problem is approximately solved by generating adversarial examples using PGD attack and then minimizing the classification loss of the adversarial example. In this paper, we propose to incorporate adversarial training in Bayesian neural network to achieve better robustness.

The other example is RSE (Liu et al., 2017), in this algorithm the authors proposed a "noise layer", which fuses input features with Gaussian noise. They show empirically that an ensemble of models can increase the robustness of deep neural networks. Besides, their method can generate an infinite number of models on-the-fly without any additional memory cost. The noise layer is applied in both training and testing phases, so the prediction accuracy will not be largely affected. Our algorithm is different from RSE in two folds: 1) We add noise to each weight instead of input or hidden feature, and formally model it as a BNN. 2) We incorporate adversarial training to further improve the performance.

## 2.2 BAYESIAN NEURAL NETWORKS (BNN)

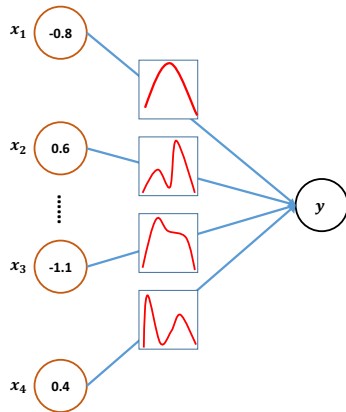

Figure 1: Illustration of Bayesian neural networks.

The idea of BNN is illustrated in Fig. 1. Given the observable random variables $(\boldsymbol{x}, y)$, we aim to estimate the distributions of hidden variables $\boldsymbol{w}$. In our case, the observable random variables correspond to the features $\boldsymbol{x}$ and labels $y$, and we are interested in the posterior over the weights $p(\boldsymbol{w}|\boldsymbol{x}, y)$ given the prior $p(\boldsymbol{w})$. However, the exact solution of posterior is often intractable: notice that $p(\boldsymbol{w}|\boldsymbol{x}, y) = \frac{p(\boldsymbol{x}, y|\boldsymbol{w})p(\boldsymbol{w})}{p(\boldsymbol{x}, y)}$ but the denominator involves a high dimensional integral (Blei et al., 2017), hence the conditional probabilities are hard to compute. To speedup inference, we generally have two approaches—we can either sample $\boldsymbol{w} \sim p(\boldsymbol{w}|\boldsymbol{x}, y)$ efficiently without knowing the closed-form formula through, for example, Stochastic Gradient Langevin Dynamics (SGLD) (Welling & Teh, 2011), or we can approximate the true posterior $p(\boldsymbol{w}|\boldsymbol{x}, y)$ by a parametric distribution $q_{\boldsymbol{\theta}}(\boldsymbol{w})$, where the unknown parameter $\boldsymbol{\theta}$ is estimated by minimizing $\mathsf{KL}\big(q_{\boldsymbol{\theta}}(\boldsymbol{w}) \,\|\, p(\boldsymbol{w}|\boldsymbol{x}, y)\big)$ over $\boldsymbol{\theta}$. For

neural network, the exact form of KL-divergence can be unobtainable, but we can easily find an unbiased gradient estimator of it using backward propagation, namely *Bayes by Backprop* (Blundell et al., 2015).

Despite that both methods are widely used and analyzed in-depth, they have some obvious shortcomings, making high dimensional Bayesian inference remain to be an open problem. For SGLD and its extension (e.g. (Li et al., 2016)), since the algorithms are essentially SGD updates with extra Gaussian noise, they are very easy to implement. However, they can only get one sample $\boldsymbol{w} \sim p(\boldsymbol{w}|\boldsymbol{x}, y)$ in each minibatch iteration at the cost of one forward-backward propagation, thus not efficient enough for fast inference. In addition, as the step size $\eta_t$ in SGLD decreases, the samples become more and more correlated so that one needs to generate many samples in order to control the variance. Conversely, the variational inference method is efficient to generate samples since we know the approximated posterior $q_{\boldsymbol{\theta}}(\boldsymbol{w})$ once we minimized the KL-divergence. The problem is that for simplicity we often assume the approximation $q_{\boldsymbol{\theta}}$ to be a fully factorized Gaussian distribution:

$$q_{\boldsymbol{\theta}}(\boldsymbol{w}) = \prod_{i=1}^{d} q_{\boldsymbol{\theta}_i}(\boldsymbol{w}_i), \text{ and } q_{\boldsymbol{\theta}_i}(\boldsymbol{w}_i) = \mathcal{N}(\boldsymbol{w}_i; \boldsymbol{\mu}_i, \boldsymbol{\sigma}_i^2). \tag{5}$$

Although our assumption (5) has a simple form, it inherits the main drawback from mean-field approximation. When the ground truth posterior has significant correlation between variables, the approximation in (5) will have a large deviation from true posterior $p(\boldsymbol{w}|\boldsymbol{x}, y)$. This is especially true for convolutional neural networks, where the values in the same convolutional kernel seem to be highly correlated. However, we still choose this family of distribution in our design as the simplicity and efficiency are mostly concerned.

In fact, there are many techniques in deep learning area borrowing the idea of Bayesian inference without mentioning explicitly. For example, Dropout (Srivastava et al., 2014) is regarded as a powerful regularization tool for deep neural networks, which applies an element-wise product of the feature maps and i.i.d. Bernoulli or Gaussian r.v. $\mathcal{B}(1, \alpha)$ (or $\mathcal{N}(1, \alpha)$). If we allow each dimension to have an independent dropout rate and take them as model parameters to be learned, then we can extend it to the variational dropout method (Kingma et al., 2015). Notably, learning the optimal dropout rates for data relieves us from manually tuning hyper-parameter on hold-out data. Similar idea is also used in RSE (Liu et al., 2017), except that it was used to improve the robustness under adversarial attacks. As we discussed in the previous section, RSE incorporates Gaussian noise $\boldsymbol{\epsilon} \sim \mathcal{N}(0, \sigma^2)$ in an additive manner, where the variance $\sigma^2$ is user predefined in order to maximize the performance. Different from RSE, our Adv-BNN has two degrees of freedom (mean and variance) and the network is trained on adversarial examples.

## 3 METHOD

In our method, we combine the idea of adversarial training (Madry et al., 2017) with Bayesian neural network, hoping that the randomness in the weights $\boldsymbol{w}$ provides stronger protection for our model.

To build our Bayesian neural network, we assume the joint distribution $q_{\boldsymbol{\mu},\boldsymbol{s}}(\boldsymbol{w})$ is fully factorizable (see (5)), and each posterior $q_{\boldsymbol{\mu}_i,\boldsymbol{s}_i}(\boldsymbol{w}_i)$ follows normal distribution with mean $\boldsymbol{\mu}_i$ and standard deviation $\exp(\boldsymbol{s}_i) > 0$. The prior distribution is simply isometric Gaussian $\mathcal{N}(\boldsymbol{0}_d, s_0^2 \boldsymbol{I}_{d \times d})$. We choose the Gaussian prior and posterior for its simplicity and closed-form KL-divergence, that is, for any two Gaussian distributions $s$ and $t$,

$$\mathsf{KL}(s \parallel t) = \log \frac{\sigma_t}{\sigma_s} + \frac{\sigma_s^2 + (\mu_s - \mu_t)^2}{2\sigma_t^2} - 0.5, \qquad s \text{ or } t \sim \mathcal{N}(\mu_{s \text{ or } t}, \sigma_{s \text{ or } t}^2). \tag{6}$$

Note that it is also possible to choose more complex priors such as "spike-and-slab" (Ishwaran et al., 2005) or Gaussian mixture, although in these cases the KL-divergence of prior and posterior is hard to compute and practically we replace it with the Monte-Carlo estimator, which has higher variance, resulting in slower convergence rate (Kingma, 2017).

Following the recipe of variational inference, we adapt the robust optimization to the evidence lower bound (ELBO) *w.r.t.* the variational parameters during training. First of all, recall the ELBO on the

original dataset (the unperturbed data) can be written as

$$- \mathsf{KL}\big(q_{\boldsymbol{\mu},\boldsymbol{s}}(\boldsymbol{w}) \parallel p(\boldsymbol{w})\big) + \sum_{(\boldsymbol{x}_i,y_i)\in\mathcal{D}_{\mathrm{tr}}} \mathbb{E}_{\boldsymbol{w}\sim q_{\boldsymbol{\mu},\boldsymbol{s}}} \log p(y_i|\boldsymbol{x}_i,\boldsymbol{w}), \tag{7}$$

rather than directly maximizing the ELBO in (7), we consider the following alternative objective,

$$\mathcal{L}(\boldsymbol{\mu},\boldsymbol{s}) \triangleq -\mathsf{KL}\big(q_{\boldsymbol{\mu},\boldsymbol{s}}(\boldsymbol{w}) \parallel p(\boldsymbol{w})\big) + \sum_{(\boldsymbol{x}_i,y_i)\in\mathcal{D}_{\mathrm{tr}}} \min_{\|\boldsymbol{x}_i^{\mathrm{adv}}-\boldsymbol{x}_i\|\leq\gamma} \mathbb{E}_{\boldsymbol{w}\sim q_{\boldsymbol{\mu},\boldsymbol{s}}} \log p(y_i|\boldsymbol{x}_i^{\mathrm{adv}},\boldsymbol{w}). \tag{8}$$

This is essentially finding the minima for each data point $(\boldsymbol{x}_i,y_i) \in \mathcal{D}_{\mathrm{tr}}$ inside the $\gamma$-norm ball, we can also interpret (8) as an even looser lower bound of evidence. So the robust optimization procedure is to maximize (8), i.e.

$$\boldsymbol{\mu}^*,\boldsymbol{s}^* = \arg\max_{\boldsymbol{\mu},\boldsymbol{s}} \mathcal{L}(\boldsymbol{\mu},\boldsymbol{s}). \tag{9}$$

To make the objective more specific, we combine (8) with (9) and get

$$\arg\max_{\boldsymbol{\mu},\boldsymbol{s}} \Big\{ \Big[ \sum_{(\boldsymbol{x}_i,y_i)\in\mathcal{D}_{\mathrm{tr}}} \min_{\|\boldsymbol{x}_i^{\mathrm{adv}}-\boldsymbol{x}_i\|\leq\gamma} \mathbb{E}_{\boldsymbol{w}\sim q_{\boldsymbol{\mu},\boldsymbol{s}}} \log p(y_i|\boldsymbol{x}_i^{\mathrm{adv}},\boldsymbol{w}) \Big] - \mathsf{KL}\big(q_{\boldsymbol{\mu},\boldsymbol{s}}(\boldsymbol{w}) \parallel p(\boldsymbol{w})\big) \Big\} \tag{10}$$

In our case, $p(y|\boldsymbol{x}^{\mathrm{adv}},\boldsymbol{w}) = \mathrm{Softmax}\big(f(\boldsymbol{x}_i^{\mathrm{adv}};\boldsymbol{w})\big)[y_i]$ is the network output on the adversarial sample $(\boldsymbol{x}_i^{\mathrm{adv}},y_i)$. More generally, we can reformulate our model as $y = f(\boldsymbol{x};\boldsymbol{w})+\zeta$ and assume the residual $\zeta$ follows either $\mathrm{Logistic}(0,1)$ or Gaussian distribution depending on the specific problem, so that our framework includes both classification and regression tasks. We can see that the only difference between our Adv-BNN and the standard BNN training is that the expectation is now taken over the adversarial examples $(\boldsymbol{x}^{\mathrm{adv}},y)$, rather than natural examples $(\boldsymbol{x},y)$. Therefore, at each iteration we first apply a randomized PGD attack (as introduced in eq (3)) for $T$ iterations to find $\boldsymbol{x}^{\mathrm{adv}}$, and then fix the $\boldsymbol{x}^{\mathrm{adv}}$ to update $\boldsymbol{\mu},\boldsymbol{s}$.

When updating $\boldsymbol{\mu}$ and $\boldsymbol{s}$, the KL term in (8) can be calculated exactly by (6), whereas the second term is very complex (for neural networks) and can only be approximated by sampling. Besides, in order to fit into the back-propagation framework, we adopt the *Bayes by Backprop* algorithm (Blundell et al., 2015). Notice that we can reparameterize $\boldsymbol{w} = \boldsymbol{\mu} + \exp(\boldsymbol{s}) \odot \boldsymbol{\epsilon}$, where $\boldsymbol{\epsilon} \sim \mathcal{N}(\boldsymbol{0}_d,\boldsymbol{I}_{d\times d})$ is a parameter free random vector, then for any differentiable function $h(\boldsymbol{w},\boldsymbol{\mu},\boldsymbol{s})$, we can show that

$$\begin{aligned} \frac{\partial}{\partial\boldsymbol{\mu}} \mathbb{E}_{\boldsymbol{w}}[h(\boldsymbol{w},\boldsymbol{\mu},\boldsymbol{s})] &= \mathbb{E}_{\boldsymbol{\epsilon}} \Big[ \frac{\partial}{\partial\boldsymbol{w}} h(\boldsymbol{w},\boldsymbol{\mu},\boldsymbol{s}) + \frac{\partial}{\partial\boldsymbol{\mu}} h(\boldsymbol{w},\boldsymbol{\mu},\boldsymbol{s}) \Big] \\ \frac{\partial}{\partial\boldsymbol{s}} \mathbb{E}_{\boldsymbol{w}}[h(\boldsymbol{w},\boldsymbol{\mu},\boldsymbol{s})] &= \mathbb{E}_{\boldsymbol{\epsilon}} \Big[ \exp(\boldsymbol{s}) \odot \boldsymbol{\epsilon} \odot \frac{\partial}{\partial\boldsymbol{w}} h(\boldsymbol{w},\boldsymbol{\mu},\boldsymbol{s}) + \frac{\partial}{\partial\boldsymbol{s}} h(\boldsymbol{w},\boldsymbol{\mu},\boldsymbol{s}) \Big]. \end{aligned} \tag{11}$$

Now the randomness is decoupled from model parameters, and thus we can generate multiple $\boldsymbol{\epsilon}$ to form a unbiased gradient estimator. To integrate into deep learning framework more easily, we also designed a new layer called `RandLayer`, which is summarized in appendix.

It is worth noting that once we assume the simple form of variational distribution (5), we can also adopt the *local reparameterization trick* (Kingma et al., 2015). That is, rather than sampling the weights $\boldsymbol{w}$, we directly sample the activations and enjoy the lower variance during the sampling process. Although in our experiments we find the simple *Bayes by Backprop* method efficient enough.

For ease of doing SGD iterations, we rewrite (9) into a finite sum problem by dividing both sides by the number of training samples $N_{\mathrm{tr}}$

$$\boldsymbol{\mu}^*,\boldsymbol{s}^* = \arg\min_{\boldsymbol{\mu},\boldsymbol{s}} \underbrace{-\frac{1}{N_{\mathrm{tr}}} \sum_{i=1}^{N_{\mathrm{tr}}} \log p(y_i|\boldsymbol{x}_i^{\mathrm{adv}},\boldsymbol{w})}_{\text{classification loss}} + \underbrace{\frac{1}{N_{\mathrm{tr}}} g(\boldsymbol{\mu},\boldsymbol{s})}_{\text{regularization}}, \tag{12}$$

here we define $g(\boldsymbol{\mu},\boldsymbol{s}) \triangleq \mathsf{KL}(q_{\boldsymbol{\mu},\boldsymbol{s}}(\boldsymbol{w}) \parallel p(\boldsymbol{w}))$ by the closed form solution (6), so there is no randomness in it. We sample new weights by $\boldsymbol{w} = \boldsymbol{\mu} + \exp(\boldsymbol{s}) \odot \boldsymbol{\epsilon}$ in each forward propagation, so that the stochastic gradient is unbiased. In practice, however, we need a weaker regularization for

small dataset or large model, since the original regularization in (12) can be too large. We fix this problem by adding a factor $0 < \alpha \leq 1$ to the regularization term, so the new loss becomes

$$-\frac{1}{N_{\text{tr}}} \sum_{i=1}^{N_{\text{tr}}} \log p(y_i | \boldsymbol{x}_i^{\text{adv}}, \boldsymbol{w}) + \frac{\alpha}{N_{\text{tr}}} g(\boldsymbol{\mu}, \boldsymbol{s}), \quad 0 < \alpha \leq 1. \tag{13}$$

In our experiments, we found little to no performance degradation compared with the same network without randomness, if we choose a suitable hyper-parameter $\alpha$, as well as the prior distribution $\mathcal{N}(\boldsymbol{0}, s_0^2 \boldsymbol{I})$.

The overall training algorithm is shown in Alg. 1. To sum up, our Adv-BNN method trains an arbitrary Bayesian neural network with the min-max robust optimization, which is similar to Madry et al. (2017). As we mentioned earlier, even though our model contains noise and eventually the gradient information is also noisy, by doing multiple forward-backward iterations, the noise will be cancelled out due to the law of large numbers. This is also the suggested way to bypass some stochastic defenses in Athalye et al. (2018).

---

**Algorithm 1** Code snippet for training Adv-BNN

---

1: **procedure** **pgd_attack**($\boldsymbol{x}, y, \boldsymbol{w}$)
2:      ▷ *Perform the PGD-attack* (2)*, omitted for brevity*
3: **procedure** **train**(data, $\boldsymbol{w}$)
4:      ▷ *Input: dataset and network weights* $\boldsymbol{w}$
5:      **for** ($\boldsymbol{x}, y$) **in** data **do**
6:          $\boldsymbol{x}^{\text{adv}} \leftarrow$ **pgd_attack**($\boldsymbol{x}, y, \boldsymbol{w}$)                       ▷ *Generate adversarial images*
7:          $\boldsymbol{w} \leftarrow \boldsymbol{\mu} + \exp(\boldsymbol{s}) \odot \boldsymbol{\epsilon}, \boldsymbol{\epsilon} \sim \mathcal{N}(\boldsymbol{0}_d, \boldsymbol{I}_{d \times d})$        ▷ *Sample new model parameters*
8:          $\hat{y} \leftarrow$ **forward**($\boldsymbol{w}, \boldsymbol{x}^{\text{adv}}$)                             ▷ *Forward propagation*
9:          loss_ce $\leftarrow$ **cross_entropy**($\hat{y}, y$)                      ▷ *Cross-entropy loss*
10:         loss_kl $\leftarrow$ **kl_divergence**($\boldsymbol{w}$)                    ▷ *KL-divergence* following (6)
11:         $\mathcal{L}(\boldsymbol{\mu}, \boldsymbol{s}) \leftarrow$ loss_ce $+ \frac{\alpha}{N_{\text{tr}}} \cdot$ loss_kl            ▷ *Total loss following* (13)
12:         $\frac{\partial \mathcal{L}}{\partial \boldsymbol{\mu}}, \frac{\partial \mathcal{L}}{\partial \boldsymbol{s}} \leftarrow$ **backward**$\big(\mathcal{L}(\boldsymbol{\mu}, \boldsymbol{s})\big)$          ▷ *Backward propagation to get gradients*
13:         $\boldsymbol{\mu}, \boldsymbol{s} \leftarrow \boldsymbol{\mu} - \eta_t \frac{\partial \mathcal{L}}{\partial \boldsymbol{\mu}}, \boldsymbol{s} - \eta_t \frac{\partial \mathcal{L}}{\partial \boldsymbol{s}}$       ▷ *SGD update, omitting momentum and weight decay*
14:      **return** net

---

Will it be beneficial to have randomness in adversarial training? After all, both randomized network and adversarial training can be viewed as different ways for controlling local Lipschitz constants of the loss surface around the image manifold, and thus it is non-trivial to see whether combining those two techniques can lead to better robustness. The connection between randomized network (in particular, RSE) and local Lipschitz regularization has been derived in Liu et al. (2017). Adversarial training can also be connected to local Lipschitz regularization with the following arguments. Recall that the loss function given data $(\boldsymbol{x}_i, y_i)$ is denoted as $\ell\big(f(\boldsymbol{x}_i; \boldsymbol{w}), y_i\big)$, and similarly the loss on perturbed data $(\boldsymbol{x}_i + \boldsymbol{\xi}, y_i)$ is $\ell\big(f(\boldsymbol{x}_i + \boldsymbol{\xi}; \boldsymbol{w}), y_i\big)$. Then if we expand the loss to the first order

$$\Delta\ell \triangleq \ell\big(f(\boldsymbol{x}_i + \boldsymbol{\xi}; \boldsymbol{w}), y_i\big) - \ell\big(f(\boldsymbol{x}_i; \boldsymbol{w}), y_i\big) = \boldsymbol{\xi}^\mathsf{T} \nabla_{\boldsymbol{x}_i} \ell\big(f(\boldsymbol{x}_i; \boldsymbol{w}), y_i\big) + \mathcal{O}(\|\boldsymbol{\xi}\|^2), \tag{14}$$

we can see that the robustness of a deep model is closely related to the gradient of the loss over the input, i.e. $\nabla_{\boldsymbol{x}_i} \ell\big(f(\boldsymbol{x}_i), y_i\big)$. If $\|\nabla_{\boldsymbol{x}_i} \ell\big(f(\boldsymbol{x}_i), y_i\big)\|$ is large, then we can find a suitable $\boldsymbol{\xi}$ such that $\Delta\ell$ is large. Under such condition, the perturbed image $\boldsymbol{x}_i + \boldsymbol{\xi}$ is very likely to be an adversarial example. It turns out that adversarial training (4) directly controls the local Lipschitz value on the **training set**, this can be seen if we combine (14) with (4)

$$\min_{\boldsymbol{w}} \ell(f(\boldsymbol{x}_i^{\text{adv}}; \boldsymbol{w}), y_i) = \min_{\boldsymbol{w}} \max_{\|\boldsymbol{\xi}\| \leq \gamma} \ell(f(\boldsymbol{x}_i + \boldsymbol{\xi}; \boldsymbol{w})$$
$$= \min_{\boldsymbol{w}} \max_{\|\boldsymbol{\xi}\| \leq \gamma} \ell(f(\boldsymbol{x}_i; \boldsymbol{w}), y_i) + \boldsymbol{\xi}^\mathsf{T} \nabla_{\boldsymbol{x}_i} \ell(f(\boldsymbol{x}_i; \boldsymbol{w}), y_i) + \mathcal{O}(\|\boldsymbol{\xi}\|^2). \tag{15}$$

Moreover, if we ignore the higher order term $\mathcal{O}(\|\boldsymbol{\xi}\|^2)$ then (15) becomes

$$\min_{\boldsymbol{w}} \ell(f(\boldsymbol{x}_i; \boldsymbol{w}), y_i) + \gamma \cdot \|\nabla_{\boldsymbol{x}_i} \ell(f(\boldsymbol{x}_i; \boldsymbol{w}), y_i)\|. \tag{16}$$

In other words, the adversarial training can be simplified to Lipschitz regularization, and if the model generalizes, the local Lipschitz value will also be small on the **test set**. Yet, as (Liu & Hsieh,

2018) indicates, for complex dataset like CIFAR-10, the local Lipschitz is still very large on **test set**, even though it is controlled on **training set**. The drawback of adversarial training motivates us to combine the randomness model with adversarial training, and we observe a significant improvement over adversarial training or `RSE` alone (see the experiment section below).

# 4 EXPERIMENTAL RESULTS

In this section, we test the performance of our robust Bayesian neural networks (`Adv-BNN`) with strong baselines on a wide variety of datasets. In essence, our method is inspired by adversarial training (Madry et al., 2017) and BNN (Blundell et al., 2015), so these two methods are natural baselines. If we see a significant improvement in adversarial robustness, then it means that randomness and robust optimization have independent contributions to defense. Additionally, we would like to compare our method with `RSE` (Liu et al., 2017), another strong defense algorithm relying on randomization. Lastly, we include the models without any defense as references. For ease of reproduction, we list the hyper-parameters in the appendix. Readers can also refer to the source code on github.

It is known that adversarial training becomes increasingly hard for high dimensional data (Schmidt et al., 2018). In addition to standard low dimensional dataset such as CIFAR-10, we also did experiments on two more challenging datasets: 1) STL-10 (Coates et al., 2011), which has 5,000 training images and 8,000 testing images. Both of them are $96 \times 96$ pixels; 2) ImageNet-143, which is a subset of ImageNet (Deng et al., 2009), and widely used in conditional GAN training (Miyato & Koyama, 2018). The dataset has 18,073 training and 7,105 testing images, and all images are $64\times64$ pixels. It is a good benchmark because it has much more classes than CIFAR-10, but is still manageable for adversarial training.

## 4.1 EVALUATING MODELS UNDER WHITE BOX $\ell_\infty$-PGD ATTACK

In the first experiment, we compare the accuracy under the white box $\ell_\infty$-PGD attack. We set the maximum $\ell_\infty$ distortion to $\gamma \in$ [0:0.07:0.005] and report the accuracy on test set. The results are shown in Fig. 2. Note that when attacking models with stochastic components, we adjust PGD accordingly as mentioned in Section 2.1. To demonstrate the relative performance more clearly, we show some numerical results in Tab. 1.

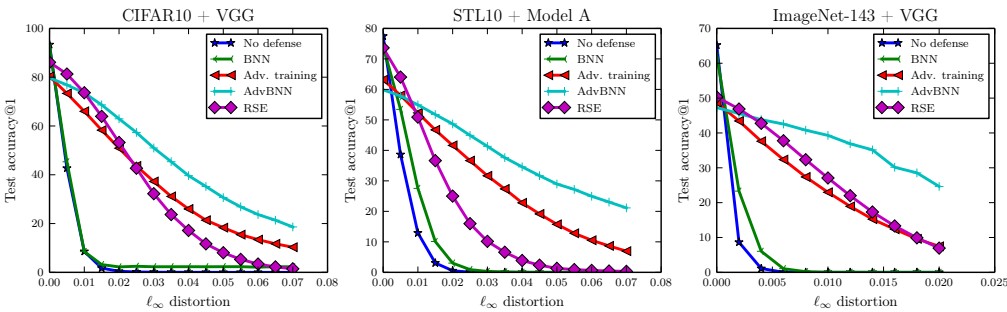

Figure 2: Accuracy under $\ell_\infty$-PGD attack on three different datasets: CIFAR-10, STL-10 and ImageNet-143. In particular, we adopt a smaller network for STL-10 namely "Model A"[1], while the other two datasets are trained on VGG.

From Fig. 2 and Tab. 1 we can observe that although BNN itself does not increase the robustness of the model, when combined with the adversarial training method, it dramatically increase the testing accuracy for $\sim 10\%$ on a variety of datasets. Moreover, the overhead of Adv-BNN over adversarial training is small: it will only double the parameter space (for storing mean and variance), and the total training time does not increase much. Finally, similar to `RSE`, modifying existing network

---

[1]Publicly available at `https://github.com/aaron-xichen/pytorch-playground/tree/master/stl10`, repository has no affiliation with us.

| Data | Defense | 0 | 0.015 | 0.035 | 0.055 | 0.07 |
|------|---------|---|-------|-------|-------|------|
| CIFAR10 | Adv. Training | **80.3** | 58.3 | 31.1 | 15.5 | 10.3 |
|  | Adv-BNN | 79.7 | **68.7** | **45.4** | **26.9** | **18.6** |
| STL10 | Adv. Training | **63.2** | 46.7 | 27.4 | 12.8 | 7.0 |
|  | Adv-BNN | 59.9 | **51.8** | **37.6** | **27.2** | **21.1** |

| Data | Defense | 0 | 0.004 | 0.01 | 0.016 | 0.02 |
|------|---------|---|-------|------|-------|------|
| ImageNet-143 | Adv. Training | **48.7** | 37.6 | 23.0 | 12.4 | 7.5 |
|  | Adv-BNN | 47.3 | **43.8** | **39.3** | **30.2** | **24.6** |

Table 1: Comparing the testing accuracy under different levels of PGD attacks. We include our method, `Adv-BNN`, and the state of the art defense method, the multi-step adversarial training proposed in Madry et al. (2017). The better accuracy is marked in **bold**. Notice that although our `Adv-BNN` incurs larger accuracy drop in the original test set (where $\|\boldsymbol{\xi}\|_\infty = 0$), we can choose a smaller $\alpha$ in (13) so that the regularization effect is weakened, in order to match the accuracy.

architectures into BNN is fairly simple, we only need to replace Conv/BatchNorm/Linear layers by their variational version. Hence we can easily build robust models based on existing ones.

## 4.2 BLACK BOX TRANSFER ATTACK

*Is our Adv-BNN model susceptible to transfer attack?* we answer this question by studying the affinity between models, because if two models are similar (e.g. in loss landscape) then we can easily attack one model using the adversarial examples crafted through the other. In this section, we measure the adversarial sample transferability between different models namely `None` (no defense), `BNN`, `Adv.Train`, `RSE` and `Adv-BNN`. This is done by the method called "transfer attack" (Liu et al., 2016). Initially it was proposed as a black box attack algorithm: when the attacker has no access to the *target model*, one can instead train a similar model from scratch (called *source model*), and then generate adversarial samples with *source model*. As we can imagine, the success rate of transfer attack is directly linked with how similar the source/target models are. In this experiment, we are interested in the following question: how easily can we transfer the adversarial examples between these five models? We study the affinity between those models, where the affinity is defined by

$$\rho_{A \mapsto B} = \frac{\text{Acc}[B] - \text{Acc}[B|A]}{\text{Acc}[B] - \text{Acc}[B|B]}, \tag{17}$$

where $\rho_{A \mapsto B}$ measures the success rate using source model $A$ and target model $B$, $\text{Acc}[B]$ denotes the accuracy of model $B$ without attack, $\text{Acc}[B|A(\text{or } B)]$ means the accuracy under adversarial samples generated by model $A(\text{or } B)$. Most of the time, it is easier to find adversarial examples through the target model itself, so we have $\text{Acc}[B|A] \geq \text{Acc}[B|B]$ and thus $0 \leq \rho_{A \mapsto B} \leq 1$. However, $\rho_{A \mapsto B} = \rho_{B \mapsto A}$ is **not** necessarily true, so the affinity matrix is not likely to be symmetric. We illustrate the result in Fig. 3.

We can observe that $\{\text{None}, \text{BNN}\}$ are similar models, their affinity is strong ($\rho \approx 0.85$) for both direction: $\rho_{\text{BNN} \mapsto \text{None}}$ and $\rho_{\text{None} \mapsto \text{BNN}}$. Likewise, $\{\text{RSE}, \text{Adv-BNN}, \text{Adv.Train}\}$ constitute the other group, yet the affinity is not very strong ($\rho \approx 0.5 \sim 0.6$), meaning these three methods are all robust to the black box attack to some extent.

## 4.3 MISCELLANEOUS EXPERIMENTS

Following experiments are not crucial in showing the success of our method, however, we still include them to help clarifying some doubts of careful readers.

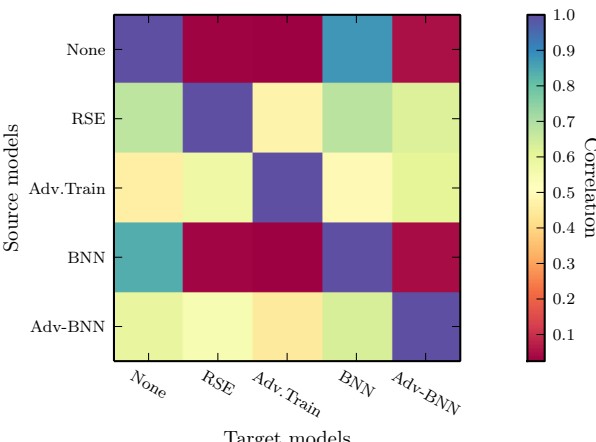

Figure 3: Black box, transfer attack experiment results. We select all combinations of source and target models trained from 5 defense methods and calculate the affinity according to (17).

The first question is about sample efficiency, recall in prediction stage we sample weights from the approximated posterior and generate the label by

$$\hat{y} = \arg\max_y \frac{1}{m} \sum_{k=1}^{m} p(y|\boldsymbol{x}, \boldsymbol{w}_k), \quad \boldsymbol{w}_k \sim q_{\boldsymbol{\mu},\boldsymbol{s}}. \tag{18}$$

In practice, we do not want to average over lots of forward propagation to control the variance, which will be much slower than other models during the prediction stage. Here we take ImageNet-143 data + VGG network as an example, to show that only 10∼20 forward operations are sufficient for robust and accurate prediction. Furthermore, the number seems to be independent on the adversarial distortion, as we can see in Fig. 4(*left*). So our algorithm is especially suitable to large scale scenario.

One might also be concerned about whether 20 steps of PGD iterations are sufficient to find adversarial examples. It has been known that for certain adversarial defense method, the effectiveness appears to be worse than claimed (Engstrom et al., 2018), if we increase the PGD-steps from 20 to 100. In Fig. 4(*right*), we show that even if we increase the number of iteration to 1000, the accuracy does not change very much. This means that even the adversary invests more resources to attack our model, its marginal benefit is negligible.

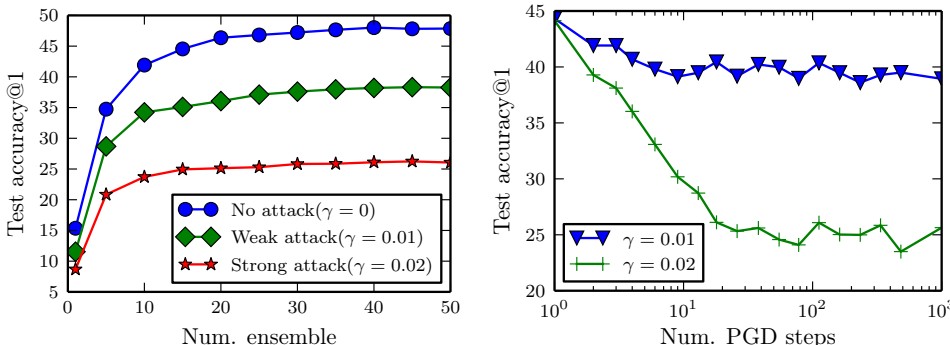

Figure 4: *Left*: we tried different number of forward propagation and averaged the results to make prediction (18). We see that for different scales of perturbation $\gamma \in \{0, 0.01, 0.02\}$, choosing number of ensemble $n = 10\sim20$ is good enough. *Right*: testing accuracy stabilizes quickly as #PGD-steps goes greater than 20, so there is no necessity to further increase the number of PGD steps.

## 5 CONCLUSION & DISCUSSION

To conclude, we find that although the Bayesian neural network has no defense functionality, when combined with adversarial training, its robustness against adversarial attack increases significantly. So this method can be regarded as a non-trivial combination of BNN and the adversarial training: robust classification relies on the controlled local Lipschitz value, while adversarial training does not generalize this property well enough to the test set; if we train the BNN with adversarial examples, the robustness increases by a large margin. Admittedly, our method is still far from the ideal case, and it is still an open problem on what the optimal defense solution will be.

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

## A   HOW TO ATTACK THE RANDOMIZED NETWORK

We largely follow the guidelines of attacking networks with "obfuscated gradients" in Athalye et al. (2018). Specifically, we derive the algorithm for white box attack to random networks denoted as $f(\boldsymbol{w}; \boldsymbol{\epsilon})$, where $\boldsymbol{w}$ is the (fixed) network parameters and $\boldsymbol{\epsilon}$ is the random vector. Many random neural networks can be reparameterized to this form, where each forward propagation returns different results. In particular, this framework includes our Adv-BNN model by setting $\boldsymbol{w} = (\boldsymbol{\mu}, \boldsymbol{s})$. Recall the prediction is made through "majority voting":

$$\hat{y} = \arg\min_y \mathbb{E}_{\boldsymbol{\epsilon}} \ell\big(f(\boldsymbol{x}; \boldsymbol{w}, \boldsymbol{\epsilon}), y\big). \tag{19}$$

So the optimal white-box attack should maximize the loss (19) on the ground truth label $y^*$. That is,

$$\boldsymbol{\xi}^* = \arg\max_{\boldsymbol{\xi}} \mathbb{E}_{\boldsymbol{\epsilon}} \ell\big(f(\boldsymbol{x} + \boldsymbol{\xi}; \boldsymbol{w}, \boldsymbol{\epsilon}), y^*\big), \tag{20}$$

and then $\boldsymbol{x}^{\text{adv}} \triangleq \boldsymbol{x} + \boldsymbol{\xi}^*$. To do that we apply SGD optimizer and sampling $\boldsymbol{\epsilon}$ at each iteration,

$$\boldsymbol{\xi}_{t+1} \leftarrow \boldsymbol{\xi}_t + \eta_t \frac{\partial}{\partial \boldsymbol{\xi}} \ell\big(f(\boldsymbol{x} + \boldsymbol{\xi}; \boldsymbol{w}, \boldsymbol{\epsilon}), y^*\big)\Big|_{\substack{\boldsymbol{\xi} = \boldsymbol{\xi}_t \\ \boldsymbol{\epsilon} = \boldsymbol{\epsilon}_t}}, \tag{21}$$

one can see the iteration (21) approximately solves (20).

## B   FORWARD & BACKWARD IN RANDLAYER

It is very easy to implement the forward & backward propagation in BNN. Here we introduce the `RandLayer` that can seamlessly integrate into major deep learning frameworks. We take PyTorch as an example, the code snippet is shown in Alg. 1.

Algorithm 1: Code snippet for implementing `RandLayer`

```python
class RandLayerFunc(Function):
    @staticmethod
    def forward(ctx, mu, sigma, eps, sigma_0, N):
        eps.normal_()
        ctx.save_for_backward(mu, sigma, eps)
        ctx.sigma_0 = sigma_0
        ctx.N = N
        return mu + torch.exp(sigma) * eps
    @staticmethod
    def backward(ctx, grad_output):
        mu, sigma, eps = ctx.saved_tensors
        sigma_0, N = ctx.sigma_0, ctx.N
        grad_mu = grad_sigma = grad_eps = grad_sigma_0 = grad_N = None
        tmp = torch.exp(sigma)
```

```
15              if ctx.needs_input_grad[0]:
16                  grad_mu = grad_output + mu/(sigma_0*sigma_0*N)
17              if ctx.needs_input_grad[1]:
18                  grad_sigma = grad_output*tmp*eps - 1 / N + tmp*tmp/(sigma_0*sigma_0*N)
19              return grad_mu, grad_sigma, grad_eps, grad_sigma_0, grad_N
20   rand_layer = RandLayerFunc.apply
```

Based on `RandLayer`, we can further implement variational `Linear` layer below in Alg. 2. The other layers such as Conv/BatchNorm are very similar.

Algorithm 2: Code snippet for implementing variational `Linear` layer

```
1    class Linear(Module):
2        def __init__(self, d_in, d_out):
3            self.d_in = d_in
4            self.d_in = d_in
5            self.d_out = d_out
6            self.init_s = init_s
7            self.mu_weight = Parameter(torch.Tensor(d_out, d_in))
8            self.sigma_weight = Parameter(torch.Tensor(d_out, d_in))
9            self.register_buffer('eps_weight', torch.Tensor(d_out, d_in))
10       def forward(self, x):
11           weight = rand_layer(self.mu_weight, self.sigma_weight, self.eps_weight)
12           bias = None
13           return F.linear(input, weight, bias)
```

## C  HYPER-PARAMETERS

We list the key hyper-parameters in Tab. 2, note that we did not tune the hyper-parameters very hard, therefore it is entirely possible to find better ones.

| Name | Value | Notes |
|------|-------|-------|
| $k$ | 20 | #PGD iterations in attack |
| $k'$ | 10 | #PGD iterations in adversarial training |
| $\gamma$ | CIFAR10/STL10: 8/256, ImageNet: 0.01 | $\ell_\infty$-norm in adversarial training |
| $\sigma_0$ | CIFAR10: 0.05, others: 0.15 | Std. of the prior distribution (not sensitive) |
| $\alpha$ | CIFAR10: 1.0, others: 1.0/50 | See (13) |
| $n$ | 10~20 | #Forward passes when doing ensemble inference |

Table 2: Hyper-parameters setting in our experiments.

