# OpenReview forum: "Adv-BNN: Improved Adversarial Defense through Robust Bayesian Neural Network"
_ICLR.cc/2019/Conference_

### Official Review · AnonReviewer1 · 2018-10-25
**A nice paper that bridges adversarial training and Bayesian neural nets**

**Rating:** 7
**Confidence:** 3

**Review:**

The paper extends the PGD adversarial training method (Madry et al., 2017) to Bayesian Neural Nets (BNNs).
The proposed method defines a generative process that ties the prediction output and the adversarial input
pattern via a set of shared neural net weights. These weights are then assinged a prior and
the resultant posterior is approximated by variational inference.

Strength:
  * The proposed approach is incremental, but anyway novel.
  * The results are groundbreaking.
  * There are some technical flaws in the way the method has been presented,
but the rest of the paper is very well-written.

Major Weaknesses:

  * Equation 7 does not seem to be precise. First, the notation p(x_adv, y | w) is severely misleading. If x_adv is also an input, no matter if stochastic or deterministic, the likelihood should read p(y | w, x_adv). Furthermore, if the resultant method is a BNN with an additional expectation on x_adv, the distribution employed on x_adv resulting from the attack generation process should also be written in the form of the related probability distribution (e.g. N(x_adv|x,\sigma)).

  * Second, the constraint that x_adv should lie within the \gamma-ball of x has some implications on the validity of
the Jensen's inequality, which relates Equation 7 to proper posterior inference.

  * Blundell et al.'s algorithm should be renamed to "Bayes-by-BACKprop". This is also an outdated inference technique for quite many scenarios including the one presented in this paper. Why did not the authors benefit from the local reparametrization trick that enjoy much lower estimator variance? There even emerge sampling-free techniques that nullify this variance altogether and provide much more stable training experience.

And Some Minor Issues:

  * The introduction part of paper is unnecessarily long and the method part is in turn too thin. As a reader, I would prefer getting deeper into the proposed method instead of reading side material which I can also find in the cited articles.

  * I do symphathize and agree that Python is a dominant language in the ML community. Yet, it is better scientific writing practice to provide language-independent algorithmic findings as pseudo-code instead of native Python.

Overall, this is a solid work with a novel method and very strong experimental findings. Having my grade discounted due to the technical issues I listed above and the limitedness of the algorithmic novelty, I still view it as an accept case.

---

> ### Author Response · Authors · 2018-11-10
> **Thanks for your helpful suggestions!**
>
> We thank the reviewer for valuing our paper and giving informative suggestions, below we address your comments in detail.
>
> Major Weaknesses:
> 1. Thanks for pointing out this mistake, this issue is also noticed by AnonReviewer 2 and we have already fixed it in the revised paper.  And perhaps it will be clearer to think our objective function as an expectation on the original data, rather than on x_adv. Because the new objective function is a lower bound of the original ELBO.
>
> 2. The evidence is not calculated on x_adv, but on the original data $x$. So it does not interfere with Jensen's ineq. when deriving the ELBO. I think it will be clearer to see the revised paper, where we give more details regarding the objective function.
>
> 3. We have renamed to "Bayes by Backprop" in the revised version.
>
> We agree with you that the local reparamterization trick has much smaller variance during the training time, replacing Bayes-by-Backprop by local reparameterization trick will definitely have a faster convergence. The reason is that we didn’t think very carefully at the implementation stage and somehow "forgot" it. Nevertheless, both algorithms should yield similar results and we will definitely try this idea and replace the code base.
>
> We want to address that our main goal is to combine Bayesian NN with adversarial training, and there are many ways we could do the approximate inference efficiently. Here we only choose a naive approach considering its simplicity and effectiveness.
>
> In the revised paper, we give readers a reminder that local reparametrization trick should perform better.
>
>
> Minor issues:
> 1. The original introduction includes intro, background and related work. We have split it into 2 sections and shortened each of them. We have also added more details in the proposed method section.
>
> 2. Thanks for your suggestion, we rewrite the algorithm box to pseudo code in order to make it looks more formal.

---

### Official Review · AnonReviewer3 · 2018-10-25
**An approach that can work well in practice, but not principled**

**Rating:** 6
**Confidence:** 4

**Review:**

I have read the feedback and discussed with the authors on my concerns for a few rounds.

The revision makes much more sense now, especially by removing section 3.3 and replacing it with more related experiments.

I have a doubt on whether the proposed method is principled (see below discussions). The authors responded honestly and came up with some other solution. A principled approach of adversarially training BNNs is still unknown, but I'm glad that the authors are happy to think about this problem.

I have raised the score to 6. I wouldn't mind seeing this paper accepted, and I believe this method as a practical solution will work well for VI-based BNNs. But again, this score "6" reflects my opinion that the approach is not principled.

=========================================================

Thank you for an interesting read.

The paper proposes training a Bayesian neural network (BNN) with adversarial training. To the best of my knowledge the idea is new (although from my perspective is quite straight-forward, but see some discussions below). The paper is well written and easy to understand. Experimental results are promising, but I don't understand how the last experiment relates to the main idea, see comments below.

There are a few issues to be addressed in revision:

1. The paper seems to have ignored many papers in BNN literature on defending adversarial attacks. See e.g. [1][2][3][4] and papers citing them. In fact robustness to adversarial attacks is becoming a standard test case for developing approximate inference on Bayesian neural networks. This means Figure 2 is misleading as in the paper "BNN" actually refers to BNN with mean-field variational Gaussian approximations.

2. Carlini and Wagner (2017a) has discussed a CW-based attack that can increase the success rate of attack on (dropout) BNNs, which can be easily transferred to a corresponding PGD version. Essentially the PGD attack tested in the paper does not assume the knowledge of BNN, let alone the adversarial training. This seems to contradict to the pledge in Athalye et al. that the defence method should be tested against an attack that is aware of the defence.

3. I am not exactly sure if equation 7 is the most appropriate way to do adversarial training for BNNs. From a modelling perspective, if we can do Bayesian inference exactly, then after marginalisation of w, the model does NOT assume independence between datapoints. This means if we want to attack the model, then we need to do
\min_{||\delta_x|| < \gamma} log p(D_adv),
D_adv = {(x + \delta_x, y) | (x, y) \sim \sim D_tr},
log p(D_adv) = \log \int \prod_{(x, y) \sim D_tr} p(y|x + \delta_x, w) p(w) dw.
Now the model evidence log p(D_adv) is intractable and you resort to variational lower-bound. But from the above equation we can see the lower bound writes as
\min_{||\delta_x|| < \gamma} \max_{q} E_{q} [\sum_{(x, y) \sim D_tr} \log p(y|x + \delta_x, w) ] - KL[q||p],
which is different from your equation 7. In fact equation 7 is a lower-bound of the above, which means the adversaries are somehow "weakened".

4. I am not exactly sure the purpose of section 3.3. True, that variational inference has been used for compressing neural networks, and the experiment in section 3.3 also support this. However, how does network pruning relate to adversarial robustness? I didn't see any discussion on this point. Therefore section 3.3 seems to be irrelevant to the paper.

Some papers on BNN's adversarial robustness:
[1] Li and Gal. Dropout Inference in Bayesian Neural Networks with Alpha-divergences. ICML 2017
[2] Feinman et al. Detecting Adversarial Samples from Artifacts. arXiv:1703.00410
[3] Louizos and Welling. Multiplicative Normalizing Flows for Variational Bayesian Neural Networks. ICML 2017
[4] Smith and Gal. Understanding Measures of Uncertainty for Adversarial Example Detection. UAI 2018

---

> ### Author Response · Authors · 2018-11-10
> **Thanks for your helpful suggestions!**
>
> Please see the revised paper as well as the change list for details, we believe that the revised paper has already addressed the issues. Below we give more details on that,
>
> 1. The references you mentioned are indeed very relevant to our topic, we discussed some of them in the introduction section. However, we still think it does not diminish the main contributions of our paper due to the following reasons:
>
> i) [1] includes one small scale experiment on MNIST dataset, the goal is to show that although the Bayesian NN is still easily “cheated” by adversarial images, the uncertainty of predictions also increases. Meaning the Bayesian NN is aware of the epistemic uncertainty. And the authors explored this nice property in adversarial detection.
>
> Similar to [1], the experiments in [3] are still small scale (MNIST/CIFAR10), although the paper shows that the Bayesian NN has stronger adversarial robustness than a plain NN, the authors also admit that “adversarial examples are harder to escape and be uncertain about in CIFAR10, due to higher dimension”. In contrast, our proposed AdvBNN has made a huge progress in adversarial robustness: the accuracy under strong adversarial attack algorithm on even more complex, high dimensional datasets is much higher than baselines (including the Bayesian NN).
>
> ii) [2] and [4] are both on adversarial detection, while our focus is the adversarial defense, these are similar topics but different scenarios.
>
> Yes, perhaps it is not very suitable to call "BNN with factorized gaussian as approximated posterior" simply as BNN, because it does not include the previous works on BNN + adversarial attacks. But it is very straightforward to extend our work to include other inference methods.
>
> I think the major contribution of our work is that we show Bayesian neural networks empower the robustness of adversarially trained neural networks. Moreover, we demonstrate that even the most simple approximate inference method can benefit a lot to model robustness, and our method scales easily to large datasets (not just MNIST).
>
>
> 2. In fact we already assumed the attacker knows the structure of BNN in our setting (using the same approach in Carlini and Wagner (2017a) and Athalye et al). We briefly mentioned this in Section 3.1 in the initial version, and we have added more details in the revised version (see Appendix). Therefore, as you can see in our Figure 2 that BNN has a very low accuracy under attack in all datasets, which does not contradict to Athalye et al. We also use the same attack (assume the adversary knows every details of model) to test the robustness for the proposed AdvBNN model. Our conclusion is that BNN itself does not help much, but using the proposed framework, one can combine the idea of BNN with adversarial training to achieve much better robustness.
>
> Athalye et al. does not negate the effectiveness of adversarial training, for detailed information, please refer to their Github page: https://github.com/anishathalye/obfuscated-gradients, there is a table comparing the performance of different methods, among them, the adversarial training (Madry et al) has a pretty good accuracy.
>
>
> 3. We are not quite sure if we understand your point, do you mean the actual objective function should be
>                                             \min_{||\delta_x||} …. \max_{q}....
> while our objective function is
>                                             \max_{q} …… \min_{||\delta_x||} ……
> and so you think Eq 7 is an lower bound of your equation?
>
> Our objective function Eq 7 is indeed a lower bound of your proposed equation, this is because we are maximizing the “worst case” evidence lower bound. So the \max_{q} should be moved to the leftmost position.
>
> In summary, in training the model, we need to do
>                                           \max_q \min_{ ||\delta_x|| < \gamma } log p(D_adv),
>                               where log p(D_adv) = \log \int \prod_{(x, y) \sim D_tr} p(y|x + \delta_x, w) p(w) dw
> This can be further simplified to our objective function.
>
> We have added more details in the revised paper to make it clearer.
>
>
> 4. Sorry about the confusion, we also think section 3.3 is diverged from the main topic, in the revised paper, we replaced this experiment with other controlled experiments. We hope these experiments can strengthen our findings.

---

> > ### Comment · AnonReviewer3 · 2018-11-20
> > **Is your method a principled way to train BNNs with adversaries?**
> >
> > I appreciate your efforts on responding my review and updating your paper. Now the extra experiments look much more relevant to the paper which is good. Still, I would like to discuss with you, on whether your method is a principled way to train BNNs with adversaries.
> >
> > Let us set aside hyper-parameter optimizations for now and assume we have selected a good prior for the weights w. In your method you only use adversarial inputs as the observation, therefore, the exact posterior is p(w |D_adv), with D_adv containing adversarial inputs crafted on all x \in D.
> >
> > Note that if we can draw samples from the exact posterior p(w | D_adv), then in principle BNN requires **no training**, and in prediction time the BNN should be robust to adversarial examples that are crafted in a similar way as D_adv. So in this idealized setting, the adversarial game cannot be played between the adversaries and the exact posterior, because the exact posterior is not obtained by optimization.
> >
> > Apparently in practice we cannot sample from the exact posterior, and VI does introduce optimization methods to approximate p(w | D_adv). I have no problem for optimizing a lower-bound, however, I doubt whether the underlying idea of your approach is principled. In other words, does your idea generalize to other BNN inference methods, e.g. message passing and SG-MCMC? Does your method still encourage nice properties of BNNs, e.g. calibrated uncertainty?
> >
> > I would like to see a discussion on this topic. Either you need to be more specific and say your method applies to VI-BNN only, or you need to justify why your approach is principled.

---

> > > ### Author Response · Authors · 2018-11-20
> > > **Reply**
> > >
> > > Thanks for introducing this question!  We haven't tried any other inference methods during the implementation stage of this paper, but we think it is possible to extend our method, Please see our responses below:
> > >
> > > 1. The adversarial dataset D_adv not only depends on the training data D, but also the posterior p(w | D). So our method should be iterative in its nature (find adversarial examples --> inference on D_adv --> find new adversarial examples ...).
> > >
> > > 2. For general inference methods that posterior is not trained (e.g. by optimization method), we may still find an iterative algorithm, as shown below:
> > >
> > > Suppose we have a "black-box" algorithm that can do inference on data D, and the posterior is p(w|D), we may return the sample distribution p*(w) by the following iterative algorithm:
> > >
> > > Input: original training dataset D
> > > 1. Initialize posterior p0(w) := p(w | D), set loop variable i = 0
> > > 2. Perturb dataset D to get D_adv := {x+eps^* | eps^* = argmin_{ ||eps||<delta } \int_w p(y | x+eps, w) * p_i(w) dw, forall x\in D}. We can simulate the integration by sampling from p_i(w)
> > > 3. Run the "black-box" inference algorithm on D_adv to get new posterior p_{i+1}(w) := p(w | D_adv)
> > > 4. Set i = i + 1
> > > 5. GoTo step 2 until p_i(w) converges ( to p*(w) ).
> > > 6. return p*(w).
> > >
> > > The above algorithm is a natural extension of Algorithm 1 in the paper, except that here we allow the use of a more general inference method and assume the attack is on the whole dataset instead of a subset.
> > >
> > > Q: Does this method still encourage nice properties of BNNs?
> > >  A: We think p*(w) should do the job. After all, both algorithms involve the adversarial game between inference method and the attacker.

---

> > > > ### Comment · AnonReviewer3 · 2018-11-22
> > > > **Yes this method sounds more aligned to Bayesian decision theory :)**
> > > >
> > > > ...although you might need some careful derivation to figure out which data to be conditioned on, how many datapoint counts in as observations (so that the uncertainty is well calibrated), etc.
> > > >
> > > > I would encourage you to work on this direction in the future, in order to have a principled method to adversarially train BNNs. The following references might be helpful for reading:
> > > >
> > > > http://proceedings.mlr.press/v15/lacoste_julien11a/lacoste_julien11a.pdf
> > > > https://arxiv.org/pdf/1805.03901.pdf

---

### Official Review · AnonReviewer2 · 2018-10-29
**Interesting contribution**

**Rating:** 7
**Confidence:** 3

**Review:**

After feedback: I would like to thank the authors for careful revision of the paper and answering and addressing most of my concerns. From the initial submission my main concern was clarity and now the paper looks much more clearer.

I believe this is a strong paper and it represents an interesting contribution for the community.

Still things to fix:
a) a dataset used in 4.2 is not stated
b) missing articles, for example, p.5 ".In practice, however, we need a weaker regularization for A small dataset or A large model"
c) upper case at the beginning of a sentence after question: p.8 "Is our Adv-BNN model susceptible to transfer attack? we answer" - "we" -> "We"
====================================================================================

The paper proposes a Bayesian neural network with adversarial training as an approach for defence against adversarial attacks.

Main pro:
It is an interesting and reasonable idea for defence against adversarial attacks to combine adversarial training and randomness in a NN (bringing randomness into a new level in the form of a BNN), which is shown to outperform both adversarial training and random NN alone.

Main con:
Clarity. The paper does not crucially lack clarity but some claims, general organisation of the paper and style of quite a few sentences can be largely improved.

In general, the paper is sound,  the main idea appears to be novel and the paper addresses the very important and relevant problem in deep learning such as defence against adversarial attacks. Writing and general presentation can be improved especially regarding Bayesian neural networks, where some clarity issues almost become quality issues. Style of some sentences can be tuned to more formal.

In details:
1. The organisation of Section 1.1 can be improved: a general concept "Attack" and specific example "PGD Attack" are on the same level of representation, while it seems more logical that "PGD Attack" should be a subsection of "Attack". And while there is a paragraph "Attack" there is no paragraph "Defence" but rather only specific examples
2. The claim “we can either sample w ∼ p(w|x, y) efficiently without knowing the closed-form formula through the method known as Stochastic Gradient Langevin Dynamics (SGLD) (Welling & Teh, 2011)” sounds like SGLD is the only sampling method for BNN, which is not true, see, e.g., Hamiltonian Monte Carlo (Neal’s PhD thesis 1994). It is better to be formulated as "through, for example, the method ..."
3. Issues regarding eq. (7):
   a) Why there is an expectation over (x, y)? There should be the joint probability of all (x, y) in the evidence.
   b) Could the authors add more details about why it is the ELBO given that it is unconventional with adversarial examples added?
   c)  It seems that it should be log p(y | x^{adv}, \omega) rather than p(x^{adv}, y | \omega).
   d) If the authors assume noise component, i.e., y = f(x; \omega) + \epsilon, then they do not need to have a compulsory Softmax layer in their network, which is important, for example, for regression models. Then the claim “our Adv-BNN method trains an arbitrary Bayesian neural network” would be more justified
4. It would make the paper more self-contained if the Bayes by Backprop algorithm would be described in more details (space can be taken from the BNN introduction). And it seems to be a typo that it is Bayes by Backprop rather than Bayes by Prop
5. There are missing citations in the text:
    a) no models from NIPS 2017 Adversarial Attack and Defence competition (Kurakin et al. 2018) are mentioned
    b) citation to justify the claim “C&W attack and PGD attack (mentioned below) have been recognized as
two state-of-the-art white-box attacks for image classification task”
    c) “we can approximate the true posterior p(w|x, y) by a parametric distribution q_θ(w), where the unknown parameter θ is estimated by minimizing KL(q_θ(w) || p(w|x, y)) over θ” - there are a variety of works in approximate inference in BNN, it would be better to cite some of them here
    d) citation to justify the claim "although in these cases the KL-divergence of prior and posterior is hard to compute and practically we replace it with the Monte-Carlo estimator, which has higher variance, resulting in slower convergence rate.”
6. The goal and result interpretation of the correlation experiment is not very clear
7. From the presentation of Figure 4 it is unclear that this is a distribution of standard deviations of approximated posterior.
8. “To sum up, our Adv-BNN method trains an arbitrary Bayesian neural network with the adversarial examples of the same model” – unclear which same model is meant
9. "Among them, there are two lines of work showing effective results on medium-sized convolutional networks (e.g., CIFAR-10)" - from this sentence it looks like CIFAR-10 is a network rather than a dataset
10. In "Notations" y introduction is missing
11. It is better to use other symbol for perturbation rather than \boldsymbol\delta since \delta is already used for the Dirac delta function
12. “via tuning the coefficient c in the composite loss function” – the coefficient c is never introduced

Minor:
1. There are a few missing articles, for example, in Notations, “In this paper, we focus on the attack under THE norm constraint…”
2. Kurakin et al. (2017) is described in the past tense whereas Carlini & Wagner (2017a) is described in the present tense
3. Inner brackets in eq. (2) are bigger than outer brackets
4. In eq. (11) $\delta$ is not bold
5. In eq. (12) it seems that the second and third terms should have “-” rather than “+”
6. Footnote in page 6 seems to be incorrectly labelled as 1 instead of 2

---

> ### Author Response · Authors · 2018-11-10
> **Thanks for your helpful suggestions!**
>
> Please see the revised paper as well as the change list for details, below we address your comments. We find your comments very informative and we absorbed most of them in the new version.
>
>
> 1. We revised Section 1.1 following your suggestions. Specifically, we merged the PGD attack into the Attack part, and we also modified the defense part in the same way.
>
> 2. Thanks for pointing out this mistake, we agree that we left HMC behind when writing the initial draft, we modified this sentence as suggested.
>
> 3. (a) We are indeed meant to it, we changed a lot to eq. (7) in response to the suggestions of all reviewers, I hope the revised version is clearer.
>     (b) We added more details why the new objective function is still an ELBO in the updated version, briefly speaking, we made a lower bound of the original ELBO, and the lower bound of ELBO is still an evidence lower bound.
>     (c) Good point, we modified the expression in the revision, thanks for pointing out.
>     (d) It is a very good suggestion, adding an error term makes our model more general to both regression and classification problems, thanks!
>
> 4. We added a brief introduction to Bayes by Backprop. The space is really limited so forgive us if you find this part hands-waving.
>
> 5. We added more citations to support our claims
>
> 6. We gave the motivation of this experiment in section 4.2.
>     The goal of this experiment is to test the robustness under black-box attack, specifically we answer the question: “How does the Adv-BNN perform under transfer attack from other models?” and the key finding is our AdvBNN model is also very robust to blackbox attack, no matter which the source model is. Blackbox defense is also a very important task because in reality, attackers may not have access to the target model.
>
> 7. We agree with Reviewer 2 that Section 3.3 is not necessary and not quite relevant to the main point of this paper, so we removed this subsection. Instead, we added two other experiments aiming at showing the sample efficiency as well as the robustness of our model.
>
> 8 ~ 12. Thanks for pointing out our mistakes, we fixed all the typos and unclear parts as suggested.
>
>
> About your minor points
> ---------------------------------
> 1, 2, 3, 4, 6: Thanks for pointing out our typos! We fixed all of them in the revised paper.
> 5: I think Eq. (12) should be the plus sign, because we are doing the Taylor expansion: f(x+\delta, w) ~ f(x)+\delta^T \nabla f(x) + ...

---

### Author Response · Authors · 2018-11-11
**Change list**

Below are the major differences in the revised paper:

1. We removed section 3.3, because we agree with Reviewer 3 that this part is less relevant to the topic. Meanwhile, two new experiments are added to show the effectiveness of our method

2. Shorten introduction part, remove unnecessary background information.

3. Added more details when deriving the ELBO, as well as our main objective function (Eq. 7,8,9)

4. We cited some relevant papers to support some claims, according to the useful suggestions of Reviewer 2. We also improved the organization of Section 1.1

5. We added more details why our adversarial attack algorithm is sound, given the randomness of BNN. This is discussed in Appendix A.

6. We replaced the python code in Algorithm 1 with pseudo code.

7. Motivations for the transfer attack experiment

8. Fixed the imprecise description in Section 4.2, this was noticed by an anonymous reader

9. Fixed many typos.

---

### Public Comment · (anonymous) · 2018-11-11
**some comments**

I had a couple comments, some positive and some negative.

On the positive side, I appreciate that the authors carefully test for convergence of PGD (as in Figure 4) and also perform investigations on the number of models needed in the ensemble. I found both of these results helpful to me and it raised my overall impression of the paper substantially. I also found it admirable that hyperparameter settings and detailed explanations of the attack, defense, etc. were included, which aids in reproducibility. I would be even happier if the authors made their model weights publicly available so that others can test the robustness claims.

In the other direction, I wanted to raise a few concerns with claims made in the paper, although I don't see these as serious issues (rather a case of uncareful writing).

At the end of Section 3 the paper claims:
"In other words, the adversarial training can be simplified to Lipschitz regularization, and if the
model generalizes, the local Lipschitz value will also be small on the test set. Yet, as (Liu & Hsieh,
2018) indicates, for complex dataset like CIFAR-10, the local Lipschitz is still very large on test set,
even though it is controlled on training set."

I'm not sure this is a correct take-away. The issue with Lipschitz regularization is not necessarily that it does not generalize to the test set, but that regularizing the Lipschitz constant *only at individual points* is not sufficient; rather, we want the Lipschitz constant to be small across an entire neighborhood of each of the train/test points. Regularizing the pointwise Lipschitz constant tends not to do this and instead tends to lead to "gradient masking" where the gradient at a given data point is uninformative due to high curvature near that point. For a stylized illustration of this, see Figure 1 of the following paper from last year's ICLR: https://arxiv.org/abs/1801.09344.

"Obviously, it is always easier to find adversarial examples through the target model itself, so we have Acc[B|A] ≥ Acc[B|B] ". This is not true, in fact if a model performs gradient masking often the best way to attack it is via transferring from a similar model that does not have such masking. It certainly does not hold mathematically that Acc[B | A] >= Acc[B | B].

---

> ### Author Response · Authors · 2018-11-12
> **Thanks for your comments!**
>
> I am very glad to see your comments, both positive and negative ones.
>
> We have uploaded the model checkpoints to the GitHub page, sorry we cannot disclose the github link but you may find it easily.
>
> As to your other comments, here are my thoughts on that:
>
> 1. Our deduction is an extremely simplified version to the robust optimization objective. As you can see, we disagree on whether regularizing just at the training points is a good approximation to Lipschitz regularization at the neighborhoods.
>
> To me, the differences between the two regularizations are just up to higher order terms. After all, recall the PGD adversarial training sets L_inf maximum distortion to ~0.03, which is very small compared to the distance between two different images and the Taylor expansion to low order terms precisely track the original objective. So yes this simplified regularizer leads to "gradient masking", but in (Liu & Hsieh, 2018) we see even the neighborhood regularizer makes small curvature very locally, it cannot guarantee a lot to the test set.
>
> So the question is ---"when we only have limited number of training samples, how to guarantee a small curvature on the whole data generating distribution? "
>
> Our claim (not this paper's claim) is that in order to guarantee the robustness on the whole distribution, both point-wise regularization and regularize across neighborhoods may not be sufficient, but of course we can argue that the latter regularization method is a better choice.
>
>
> 2. Sorry about the confusion, in fact, the Acc[B | B] denotes the accuracy of attacking model B with model B, both models have the same architecture and *weights*. So technically it belongs to the white-box attack. We see even if the model B performs "gradient masking", if we know everything inside the model, we can still easily attack it. That's why Acc[B|A] >= Acc[B | B] is always true, because white-box attack is the strongest.
>
> We are aware that traditionally, Acc[B | B] should assume we only know the architecture but not weights, the reason we made such adaption is that we want to guarantee the relation above, and therefore a valid correlation measure (0<= \rho <= 1).

---

> > ### Public Comment · (anonymous) · 2018-11-13
> > **response**
> >
> > Re: 2, my point was not that you should not assume you know the weights. Rather, PGD is only an approximation to the best attack. If you actually had the worst-case attack then I agree that Acc[B|A] >= Acc[B | B] but given that you are making an approximation this need not hold.
> >
> > Best,
> > Same commenter as above

---

> > > ### Author Response · Authors · 2018-11-13
> > > **Reply**
> > >
> > > Yes, strictly speaking the inequality may not hold for PGD attack, because it is not guaranteed to find the optimal adversarial perturbation. Thanks for your reminding!
> > >
> > > Although in our experiments on five models (no-defense, BNN, Adv. Training, RSE, Adv-BNN), we did not observe any violation of this inequality (as you can see in Fig. 3, all correlations are within range [0, 1]).
> > >
> > > In the latest revision, we fixed this problem by changing "correlation" measure to "affinity" measure, and corrected the imprecise sentences.
> > >
> > > Again, thanks for catching this mistake!

---

### Public Comment · ~Zhanxing_Zhu1 · 2018-12-08
**It might be interesting to have a discussion about our NeurIPS2018 work "Bayesian Adversarial Learning"**

Our NeurIPS2018 work "Bayesian Adversarial Learning" also approaches the adversarial training from a Bayesian perspective, where MCMC is employed for sampling both adversarial examples and the parameters of the classifier network. Bayesian Adversarial Learning is a general framework for improving robustness of neural network to the adversarial examples. One special case of our framework is Bayesian Neural Network combined with adversarial training, when the "point" estimate of adversarial examples is used. Therefore, I think it might be interesting to have a discussion about our work.

N. Ye and Z. Zhu. "Bayesian Adversarial Learning" NeurIPS2018.

---

> ### Author Response · Authors · 2018-12-10
> **Thanks for introducing your work!**
>
> Thank you very much for introducing your recent paper on this topic! Since the paper is available after the ICLR submission deadline, we were not aware of this work. We will include some discussions and comparisons in our paper:
> - Based on our understanding, although both papers use Bayesian method to defense, the algorithms are quite different: your algorithm contains two separate SGLD sampling procedures to sample both adversarial samples and model weights, while we do not sample the adversarial samples. Instead, we integrate the adversarial training process into a single min-max optimization problem.
> - Our method (in the current form) is using variational Bayes and it makes adversarial training process much more efficient. In fact, our algorithm has time complexity similar to the original adversarial training. This can also be observed from experimental results: we are able to scale to complex datasets like CIFAR or even ImageNet-143. We are curious about how your algorithm perform under such situation and will conduct some comparisons.
> - Lastly, it seems that your paper/code are publicly available after October 26 and our submission is on September 27, so we couldn’t include the comparison/discussion in our submission. But we will definitely add discussions/comparisons into our final version.

---

### Meta-Review · Area_Chair1 · 2018-12-14

**Confidence:** 4
**Recommendation:** Accept (Poster)

**Metareview:**

Reviewers are in a consensus and recommended to accept after engaging with the authors. Please take reviewers' comments into consideration to improve your submission for the camera ready.